# A compact, high-purity source of HONO validated by Fourier Transform Infrared and Thermal Dissociation Cavity Ring-down Spectroscopy

Nicholas J. Gingerysty[1] and Hans D. Osthoff[1]

[1] Department of Chemistry, University of Calgary, 2500 University Drive N.W., Calgary, Alberta, Canada T2N 1N4

*Correspondence to*: Hans D. Osthoff (hosthoff@ucalgary.ca)

**Abstract**

A well-characterized source of nitrous acid vapour (HONO) is essential for accurate ambient air measurements by instruments requiring external calibration. In this work, a compact HONO source is described in which gas streams containing dilute concentrations of HONO are generated by flowing
hydrochloric acid (HCl) vapour emanating from a permeation tube over continuously agitated dry sodium nitrite ($NaNO_2$) heated to 50 °C. Mixing ratios of HONO and potential by-products including NO, $NO_2$ and nitrosyl chloride (ClNO) were quantified by Fourier Transform Infrared (FTIR) and thermal dissociation cavity ring-down spectroscopy (TD-CRDS). A key parameter is the concentration of HCl, which needs to be kept small (< 4 ppmv) to avoid ClNO formation. The source produces gas streams containing HONO in
air in > 95% purity relative to other nitrogen oxides. The source output is rapidly tuneable and stabilizes within 90 min. Combined with its small size and portability this source is highly suitable for calibration of HONO instruments in the field.

## 1 Introduction

The generation of nitrous acid vapour ($HONO_{(g)}$) free of other nitrogen oxides has been a long-standing challenge to atmospheric chemists (Table 1). Such sources are needed in kinetic studies, for absorption cross-section measurements, and for the calibration of field instruments. Nitrous acid is not suitable for permeation devices since HONO is difficult to prepare in low concentration and in high purity and disproportionates to nitric oxide and nitrogen dioxide via equilibrium (1).


$$NO_{(g)} + NO_{2\,(g)} + H_2O_{(g)} \rightleftharpoons 2HONO_{(g)} \tag{R1, R-1}$$

In fact, early experiments relied on the NO, $NO_2$ and $H_2O$ vapour equilibrium to form HONO in situ (King and Moule, 1962; Cox, 1974; Stockwell and Calvert, 1978).

Cox and Derwent prepared HONO vapours in ~50% purity by flowing nitrogen over an aqueous solution containing 0.1 M sodium nitrite with and 1.4%-2.5% sulfuric acid (Cox and Derwent, 1976).


$$H_2SO_{4\,(aq)} + 2NaNO_{2\,(aq)} \rightarrow 2HONO_{(g)} + 2NaHSO_{4\,(aq)} \tag{R2}$$

Braman and de la Cantera sublimed oxalic acid onto solid sodium nitrite and were able to produce gas streams in >50% and up to 90% purity as long as water was present (Braman and De la Cantera, 1986).

$$H_2C_2O_{4\,(g)} + NaNO_{2\,(s)} \xrightarrow{H_2O} HONO_{(g)} + NaHC_2O_{4\,(s)} \tag{R3}$$

Subsequent refinements included a flow reactor design in solutions of sulfuric acid and sodium nitrite were
dynamically mixed, producing a stable output in >90% purity (Taira and Kanda, 1990).

Febo et al. (1995) developed a method in which gas-phase hydrochloric acid emitted from a permeation device was quantitatively reacted in a humidified gas stream with solid sodium nitrite, producing a stable and highly pure (>99.5%) HONO output.

$$HCl_{(g)} + NaNO_{2\,(s)} \xrightarrow{H_2O} HONO_{(g)} + NaCl_{(s)} \tag{R4}$$

The output of sources based on R4 has been characterized by differential optical absorption spectroscopy (DOAS) (Febo et al., 1995; Stutz et al., 2000), Fourier transform infrared (FTIR) spectroscopy (Brust et al., 2000; Schiller et al., 2001), tuneable diode laser absorption spectroscopy (TLDAS) (Schiller et al., 2001), incoherent cavity-enhanced absorption spectroscopy (IBBCEAS) (Roberts et al., 2010), and long-path optical absorption photometry (LOPAP) (Ren et al., 2010). Stutz et al. (2000) noted that concentrations of

HONO should be kept low ($< 10^{14}$ molecules cm$^{-3}$; ~4 ppmv) to avoid disproportionation of HONO via R-1.

Perez et al. (2007) examined a source based on the design by Febo et al. (1995) using thermal-dissociation chemiluminescence (TD-CL) detection of NO and reported that their source co-emitted nitrosyl chloride (ClNO); they opted to generate HONO from reaction of $H_2SO_4$ aerosol with $NaNO_2$(s) instead. In contrast,

Roberts et al. (2010) observed little or no production of ClNO (<4.5%) and quantitative conversion of HCl to HONO, at odds with Perez et al. (2007) but in agreement with the original work by Febo et al. (1995). As far as we know, this inconsistency is unexplained to date.

As part of our development of an IBBCEAS for HONO measurement, our lab recently constructed a HONO source using an aqueous solution of $NaNO_2$ buffered with oxalate to pH 3.74 (Jordan and Osthoff, 2020).

This source generated HONO in trace amounts suitable to our needs but co-emitted NO and $NO_2$ in similar ratios as reported by Braman and De la Cantera (1986). Furthermore, this type of source could not be turned off (i.e., had to remain under continuous $N_2$ flow) and required long times to stabilize, motivating us to develop an alternative HONO generation method. Ambient air HONO measurement techniques that need to be externally calibrated often poorly agree with each other (Crilley et al., 2019), further motivating the

development of a well-characterized, compact and portable HONO source and to improve the understanding of conditions needed to suppress formation of undesired side products such as ClNO.

In this work, we describe the implementation of a compact, high-purity HONO generation device based on R4 for field deployment and laboratory experiments and characterized its output by FTIR and thermal dissociation cavity ring-down spectroscopy (TD-CRDS). Conditions to avoid generation of impurities such

as ClNO are identified. The performance of this source in comparison to existing methods is discussed.

## 2 Materials and methods

### 2.1. Generation of gas streams containing HONO

A practical challenge is the generation of stable and dilute gas streams containing HCl. Calibration gas

cylinders are the most straightforward method to use for this purpose but may require periods of up to 10 days to stabilize (Roberts et al., 2010). Furthermore, low-concentration HCl gas cylinders are expensive to source, such that we chose to generate HCl gas through the use of relatively inexpensive permeation devices.

In the setup by Febo et al. (1995), which was also implemented by Brust et al. (2000), Stutz et al. (2000)
and Ren et al. (2010), a gas stream of $N_2$ is passed through Teflon tubing immersed in an HCl bath. This
setup involved a rather large 1 L vessel containing liquid HCl which is somewhat impractical and potentially
hazardous in a field setting. With portability in mind, we decided to construct a permeation tube containing
HCl(l). Literature does not inform as to suitable tube dimensions, which were determined through trial-and-
error.

Three HONO sources were constructed and evaluated. In the first ("source 1"), a 24 cm long,
polytetrafluoroethylene (PTFE) permeation tube (Chromatographic Specialties C111LE; wall thickness
1.5 mm; outer diameter (o.d.) 7.8 mm) was filled with 2.5 mL of 37% HCl (Sigma-Aldrich) and sealed at
both ends with PTFE plugs held in place by stainless steel compression rings. The permeation tube was
placed inside a glass chamber (VICI Dynacalibrator Model 120) whose temperature was controlled at
25.0 °C and which was continuously flushed at a flow rate of ~0.15 L min$^{-1}$ with room air scrubbed using
activated charcoal for a period of several days into a fume hood. The use of scrubbed air ensured that the
gas stream contained water to maintain efficient HONO production (Schiller et al., 2001). For the
experiments shown in this manuscript, the relative humidity of the diluent gas stream was in the range of
15% to 35%.

The HCl output was diluted in a gas stream of $O_2$ (~20 mL min$^{-1}$); this gas stream also served to
continuously flush the connecting tubing when not in use. Roughly 1.5 g of solid $NaNO_2$ was placed inside
a 50 cm long loop of 3/16" (0.476 cm) inner diameter (i.d.) and ¼" (0.635 cm) o.d. fluorinated ethylene
propylene (FEP) Teflon tubing downstream from the HCl addition point. This section could be manually
bypassed with a pair of 3-way valves (Entegris) to turn HONO production on or off. A Teflon filter (Pall,
2 µm pore size and 47 mm diameter) inside a Teflon filter holder (Cole-Parmer) was placed downstream of
the $NaNO_2$ powder.

In the second source ("source 2"), the $NaNO_2$ was placed in a two-neck, 50 mL Pyrex round bottom flask
which was covered in aluminium foil to prevent photolysis of nitrite and nitrous acid and was externally
heated to a temperature of 50 °C using a water bath and mechanically agitated using a magnetic stir bar as
described by Febo et al. (1995). Heating this vessel promotes dissociation of molecular clusters such as
$N_2O_4$ and partitioning of wall-adsorbed molecules to the gas-phase, which, if present, can drive HONO
decomposition on borosilicate glass surfaces (Syomin and Finlayson-Pitts, 2003).

The third source ("optimized source") used the same setup as source 2, except that a shorter permeation tube (length 6 cm) containing 0.35 mL of HCl was used. This setup is depicted in Figure 1. The overall dimensions are 28 cm × 28 cm × 36 cm and its modular design ensures easy transport to and from the field.

To deliver HONO in atmospheric concentrations (i.e., < 10 ppbv), a portion of the source output was directed towards waste with the aid of a pump and a needle valve. The output concentration could be rapidly changed (typically by factors between 5 to 200) by adjusting the position of the needle valve. The remaining output was diluted using scrubbed air and directed towards the instruments at a final flow rate slightly larger than the sampling requirement of the instruments.

**2.2 Analysis of HONO source output by FTIR**

Gas streams exiting the HONO sources (prior to dilution) were analysed using an FTIR spectrometer (Bruker Tensor 27) equipped with a liquid nitrogen cooled mercury cadmium telluride (MCT) detector and a White multi-pass gas cell with a 6.4 m optical path length and an internal volume of 0.75 L (Gemini Scientific Instruments, Venus series) in a similar fashion as described earlier (Taha et al., 2013). Room-temperature spectra were acquired continuously at a time resolution of 30 s. Background spectra were recorded with HCl inline but with the $NaNO_2$ bypassed, which resulted in the observed spectra showing the change in trace gas concentrations.

Mixing ratios of trace gases were determined from fits (by least squares error minimization over a selected wavelength range) of room-temperature reference spectra from the Pacific Northwest National Laboratory (Sharpe et al., 2004), multiplied by the respective mixing ratios as variables, to the observed spectra. The reference spectra are provided in units of ppmv at atmospheric pressure per meter of optical length, necessitating a correction factor of 6.4 accounting for the actual path length and a pressure correction factor, for which the pressure inside the multi-pass cell (which was equal to that of the room) was monitored using a pressure transducer (Omegadyne PX419–015A5V).

The FTIR limits of detection (LODs) are specific to each molecule (due to differing absorption cross-sections) and are in the 100 to 300 parts-per-billion (by volume; ppbv; $10^{-9}$) range. For example, the $1\sigma$ precision of $NO_2$ data was ±100 ppbv, yielding a $3\sigma$ LOD of 300 ppbv.

The spectral resolution of the FTIR was ~0.5 cm$^{-1}$, which does not suffice to fully resolve the absorption lines of HCl, $H_2O$ and NO; their FTIR derived mixing ratios are hence lower limits.

### 2.3 Analysis of HONO source output by TD-CRDS

The (diluted) HONO source output was also analysed using a four-channel TD-CRDS instrument (Odame-Ankrah, 2015). Briefly, mixing ratios of $NO_2$ are quantified by absorption at 405 nm. Ring-down time constants in the absence ($\tau_0$) and presence ($\tau$) of $NO_2$ are converted to concentrations ($N$) using equation (2) where $c$ is the speed of light, $\sigma$ is the $NO_2$ absorption cross-section, and $R_L$ is a correction factor (~1.2) accounting for mirror purge flows (Paul and Osthoff, 2010).

$$N = \frac{R_L}{c\sigma}\left(\frac{1}{\tau} - \frac{1}{\tau_0}\right) \tag{1}$$

Concentrations are converted to mixing ratios using the ideal gas law. Other $NO_y$ components are converted to $NO_2$ in separate channels and are quantified by difference.

For this study, the instrument was operated as follows: Mixing ratios of $NO_2$ were monitored using a room temperature, 1/4" (0.64 cm) o.d. and 1/8" (0.32 cm) i.d. FEP Teflon inlet. Mixing ratios of $NO_x$ were quantified on a second channel with a room temperature Teflon inlet and by adding $O_3$ (mixing ratio after addition ~3 ppmv) to titrate NO to $NO_2$ (Odame-Ankrah, 2015; Fuchs et al., 2009). A third channel was operated with a 1/4" (0.64 cm) o.d. quartz inlet heated to 600 °C. Ozone was added between the quartz inlet and the CRDS cell to quantify $NO_y$, including HONO (Jordan and Osthoff, 2020; Wild et al., 2014; Perez et al., 2007). The fourth inlet was operated at 350 °C and with added $O_3$. This channel quantified $NO_x + \Sigma PAN + \Sigma AN + ClNO + ClNO_2$ and was used as a HONO reference channel. Each channel sampled at a flow rate of ~0.8 slpm through ~5 cm short, 1/16" i.d. Teflon flow restrictors placed inline after the heated quartz sections (if existing) and before a 50 mm Teflon (Pall Teflo, 2 µm pore size) housed in a Teflon filter holder (Cole Parmer). The laser pulse repetition rate was 1500 Hz, and 1500 ring-down events were averaged to produce 1 s data.

### 2.4 Field deployment

The HONO source was utilized during the "Study of nitrogen oxides in winter downwind from oil and gas sands" (SNOWDOGS) field campaign in Fort McKay, Alberta, Canada, in January 2020. Its output was quantified in parallel by TD-CRDS and a Thermo 42i-Y NO-NOy CL instrument equipped with a Mo converter heated to 325 °C. This converter and the TD-CRDS quartz inlets were mounted on the roof of a trailer which housed the instruments and the HONO source. The TD-CRDS sampled at a flow rate of ~1.2 slpm per channel through ~5 cm short, 300 µm i.d. stainless steel flow restrictors placed inline after the heated quartz sections and before the Teflon filter assembly. The HONO source output was delivered via a 5 m long, 1/4" (0.64 cm) o.d. and 3/16" (0.48 cm) i.d. FEP Teflon tube to both instruments.

## 3 Results

### 3.1 Development of a compact, high-purity HONO source

The first design tested (source 1) produced an output of ~38 ppmv of HONO from a similar amount (>38 ppmv) of HCl (Figure S1A). This HONO mixing ratio is a factor of ~10 larger than the maximum recommended by Stutz et al. (2000) to avoid partial conversion of HONO to $NO_2$ and NO via R-1. Consistently, the source output contained ~4.0 ppmv of $NO_2$ (Figure S1A) and an analogous amount of NO (Figure S1B). Furthermore, the spectrum contained an absorption feature around 1808 $cm^{-1}$, reproduced by ~15.5 ppmv of ClNO.

Nitrosyl chloride can be produced by reaction of HONO with gas-phase or wall-adsorbed HCl (Zhang et al., 1996; Wingen et al., 2000) or by reaction of two equivalents of $NO_2$ with particulate phase $Cl^-$ (Weis and Ewing, 1999), though the relatively low mixing ratios of $NO_2$ in this system makes this reaction less likely.

$$HONO_{(g)} + HCl_{(g)} \rightarrow ClNO_{(g)} + H_2O_{(g)} \tag{R5}$$

$$2NO_{2(g)} + Cl^-_{(p)} \rightarrow ClNO_{(g)} + NO_3^-_{(p)} \tag{R6}$$

The FTIR spectrum contained ~50 ppmv more $H_2O$ than the reference spectrum, i.e., more moisture than emitted by the permeation tube with $NaNO_2$ bypassed. While this is consistent with ClNO production via R5, the $H_2O$ data are likely not meaningful since the $H_2O$ mixing ratios changed slowly over time (data not shown), possibly because of slow equilibration with the inner walls of the tubing and multi-pass cell and perhaps also because of water's presence in the FTIR optical path outside the multi-pass cell at a concentration that may have drifted.

Febo et al. (1995) reported lower production of $NO_x$ by mechanically agitating the $NaNO_2$ using a stir bar (to break up pockets of high [HONO]) and by heating the reaction vessel to 50 °C; these steps were incorporated in the second design (source 2). The FTIR spectrum of its output showed a consumption of > ~42 ppmv HCl and production of ~48 ppmv HONO, ~8.0 ppmv of $NO_2$, ~6.5 ppmv of ClNO, ~8 ppmv of NO, and ~50 ppmv of $H_2O$ (Figure S2), an improvement over source 1 in terms of unwanted ClNO production but still inadequate.

A shorter HCl permeation tube was used in the third design, with the expectation that the lower HCl concentrations results in reduced generation of side products via R5 and R6. A sample FTIR spectrum of this source's output is shown in Figure 2. With a freshly prepared HCl permeation tube and an estimated

relative humidity in the round bottom flask of ~25%, > ~2.5 ppmv HCl were consumed and ~3.0 ppmv of HONO were produced. The mixing ratios of undesired side products (i.e., NO, $NO_2$, ClNO, and $HNO_3$) were below their respective FTIR detection limits (Figures 2, S3 and S4).

**3.2 Analysis of source output by TD-CRDS**

To better constrain the mole fractions of impurities, the source output was analysed by TD-CRDS. Figure 3
shows an example experiment. The TD-CRDS sampled scrubbed air before 21:38 and after 21:58 to determine the ring-down time constants in the absence of absorbers, $\tau_0$. In the time periods in between, varying amounts of the HONO output were sampled. There was little response other than in the $NO_y$ channel, the only channel sensitive to HONO. Scatter plots of $NO_2$, $NO_x$ and $NO_x + \Sigma PAN + \Sigma AN + ClNO + ClNO_2$ ("HONO ref") against $NO_y$ (Figure 3, inserts) have slopes of
(1.29±0.06)%, (1.54±0.06)% and (2.28±0.04)%, respectively, from which a source purity of >97.3% was deduced.

When concentrations changed, the 90%-10% rise (and fall) times of the TD-CRDS was < 3 s, a time constant identical to changes in $NO_x$ concentration and indicating a "well-behaved" inlet, i.e., the absence of inlet memory effects and fast equilibration with the inner walls of the inlet. The fast response contrasts
with the rise (and fall) times of $HNO_3$ of > 180 s (data not shown). Furthermore, the response in a channel sampling from an inlet heated to 600 °C without added $O_3$ was the same as the that of the $NO_2$ channel (data not shown), consistent with the absence of $HNO_3$ observed by FTIR spectroscopy (Figure S4).

A quartz inlet temperature scan when the TD-CRDS was sampling a constant concentration of HONO is shown in Figure 4. The experimental TD curve was reproduced by a fit to equation (2), which is based on
equation (4) by Paul et al. (2009).

$$[\text{Observed}]_{\text{total}} = [NO_x] + [ClNO]\left(1-e^{-A_{ClNO}\times e^{\frac{-E_{A,ClNO}}{RT}}\times t_{res}}\right) + [HONO]\left(1-e^{-A_{HONO}\times e^{\frac{-E_{A,HONO}}{RT}}\times t_{res}}\right) \quad (2)$$

Here, $A_{HONO}$ and $E_{A,HONO}$ are Arrhenius parameters for the TD of HONO taken from Tsang (1991), $A_{ClNO}$ and $E_{A,ClNO}$ are from Baulch et al. (1981), and $t_{res}$ is the residence time of the gas in the converter at the converter set temperature T. A fit of equation (2) to the observation (superimposed as a black line in Figure
4) using the software package Igor Pro (Wavemetrics) gave $t_{res}$ = (77.6±0.1) ms, $[NO_x]$ = (0.03±0.30) ppbv, $[ClNO]$ = (0.27±0.34) ppbv, and $[HONO]$ = (49.97±0.18) ppbv, suggesting a source purity of (99.4±0.9)%.

In our experience, the TD inflection points vary between TD-CRDS channels (and depend on sample flow rates), captured by the $t_{res}$ parameter that is specific to each channel. For the data shown in Figure 4, HONO

fully dissociates at a temperature of ~570 °C, and the inflection point is at ~500 °C. Nitrosyl chloride is
predicted to fully dissociate at ~330 °C (inflection point at ~288 °C), such that the temperature of 350 °C
chosen for the HONO ref channel (Figure 3) is justified. The blue, dashed line superimposed in Figure 4
shows a simulation of 5 ppbv ClNO and 45 ppbv HONO to demonstrate the relative positions of the TD of
ClNO and HONO. We attempted to verify the ClNO TD profile experimentally, but the unoptimized
source's ClNO outputs were not sufficiently stable, and those attempts were abandoned. Regardless, it is
clear from Figure 4 that the contribution of ClNO to the total source output is negligible.

As can be seen from the insert in Figure 4, HONO contributes a tiny amount to the TD-CRDS signal at
350 °C (~4 pptv in the simulation of 50 ppbv HONO). This contribution may differ between quartz heaters
and channels and could have impacted by TD-CRDS using multiple parallel channels presented at the
beginning of this section and in Figure 3, leading to an overestimation of the total amount of impurities
present in that experiment. The temperature scan data shown in Figure 4, on the other hand, were collected
using a single quartz heater and are thus more accurate.

### 3.3 Source stability

The source output gradually decreased over a time scale of weeks of continuous use, which was rationalized
by the visible depletion of the HCl permeation tube. However, the source output remained stable and
reproducible on shorter time scales. An example FTIR time series is shown in Figure 5, which was acquired
after 1 month of intermittent use. After a 1.5 hours stabilization period, the source produced a stable output
of 1.57 ppmv of HONO (from >1.0 ppmv of HCl) with a precision of ±35 ppbv.

### 3.4 Source purity and day-to-day reproducibility

The optimized source routinely delivered HONO in high purity (range 96.0% - 98.7%), as long as scrubbed,
moisture-containing room air was used as diluent gas (Table 2). When dry cylinder gas was used instead,
the source output was stable but contained a larger amount of undesired side products; for example, on July
25, 2019, the output contained 4.6% $NO_2$ and 8.8% NO + ClNO (Table 2). When used in the field, the
performance was similar: with scrubbed air, the HONO purity was 98.5%, whereas with dry cylinder gas,
the purity was merely 81.3% (Table 2 and Figure S5).

## 4. Discussion

In this work, a small footprint, portable, stable, rapidly tuneable, and high-purity (>97% on average) HONO source has been described. The source achieves compactness by using a diffusion tube containing 0.35 mL concentrated HCl rather than a 1 L HCl bath. The latter is a considerable hazard since an HCl spill can cause severe burns of skin or permanent eye damage, and hydrochloric acid vapours are toxic to inhale. Hence, smaller volumes are preferred when deploying to the field. The source stabilizes within a time frame of 1.5 h (Figure 5) which is fast compared to, for example, the 10 day stabilization period reported by Roberts et al. (2010). The source output is also rapidly tuneable (Figure 3) which allows expedient generation of calibration plots. These properties make it a useful setup for in-field instrument calibrations.

The combination of FTIR and TD-CRDS provided a unique and powerful tool set to analyse the purity of the source's output. This work has shown that HONO sources based on R4 can generate NO, $NO_2$ and ClNO as by-products via R-1 and R5. Febo et al. (1995) and Roberts et al. (2010) have presented HONO sources based on R4 and demonstrated stoichiometric conversion of HCl to HONO. The results in this work suggest that devices relying solely on loss of HCl for calibration of HONO output should be used with caution as co-production of NO, $NO_2$, and ClNO is a real possibility. In our setup, formation of ClNO could be avoided by keeping the HCl concentration low (i.e., < 4 ppmv), which puts an important constraint on sources generating HONO by R4. With this in mind, it is instructive to scrutinize the earlier work in regard to side product formation. It has generally been noted that the presence of moisture is needed to ensure a high HONO output. We suggest that water is needed in part because ClNO is prone to hydrolysis (Karlsson and Ljungstrom, 1996) such that the product distribution is shifted towards HONO.

$$ClNO_{(g)} + H_2O_{(g)} \rightarrow HONO_{(g)} + HCl_{(g)} \tag{R-5}$$

Roberts et al. (2010) observed no ClNO production, likely because they kept their concentrations relatively low (<900 ppbv) and humidified their gas streams, though they appear to be just below the threshold of possible ClNO production. Perez et al. (2007), on the other hand, observed ClNO but did not disclose details as to how they operated their source, i.e., if they worked under trace conditions and with humidified gases, though it is likely that either or both conditions were not met. Schiller et al. (2001) bubbled $N_2$ through liquid HCl and would have certainly achieved high enough HCl concentration to generate ClNO with their setup. However, they reported achieving higher HONO output in the presence of moisture. Hydrolysis of ClNO (R-5) may be a (partial) explanation why the presence of $H_2O$ improved the output of their HONO.

The present HONO source has a large dynamic range (0 - 1 ppmv), well above the HONO concentrations found in ambient air. For field calibrations in future, it may be desirable to construct a permeation tube with lower HCl output, for example by lowering the HCl concentration in the permeation tube.

## Data availability

The data used in this study are available from the corresponding author upon request
(hosthoff@ucalgary.ca).

## Author contributions

NJG and HDO designed the experiments and carried them out.

## Competing interests

The authors declare that they have no conflict of interest.

## Acknowledgments

This work was made possible by the financial support of the Natural Sciences and Engineering Research Council of Canada (NSERC) in the form of a Discovery grant to HDO (RGPIN/03849-2016). NJG acknowledges Alberta Graduate Excellence Scholarship (AGES).

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

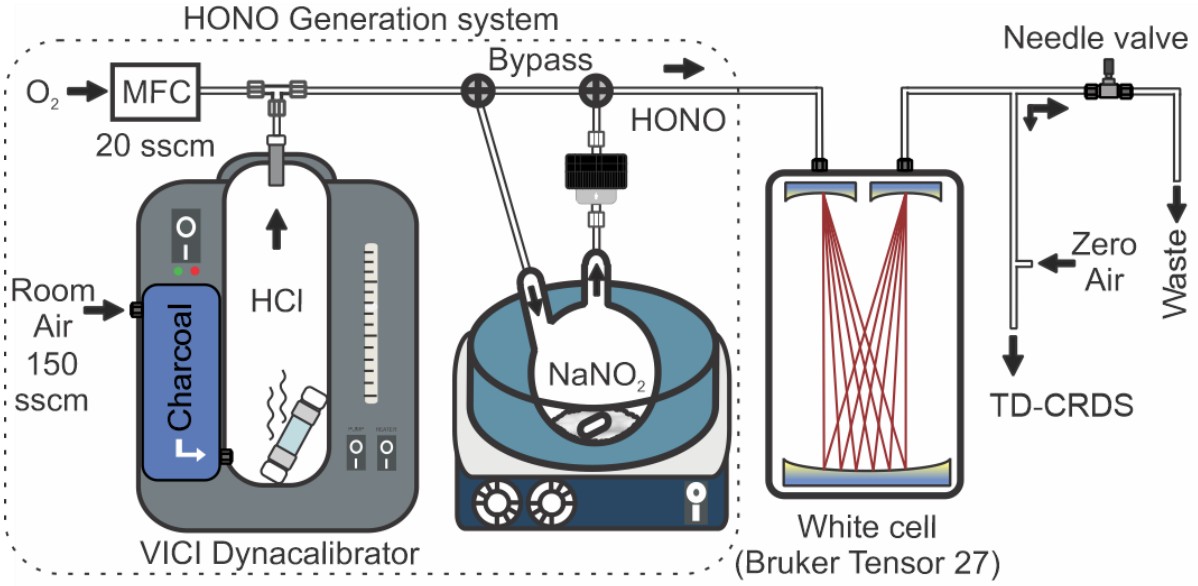

**Figure 1.** Schematics of the experimental setup. The high-purity HONO generation system is shown on the
left (dotted border). MFC = mass flow controller. sccm = standard cubic centimetre per minute.

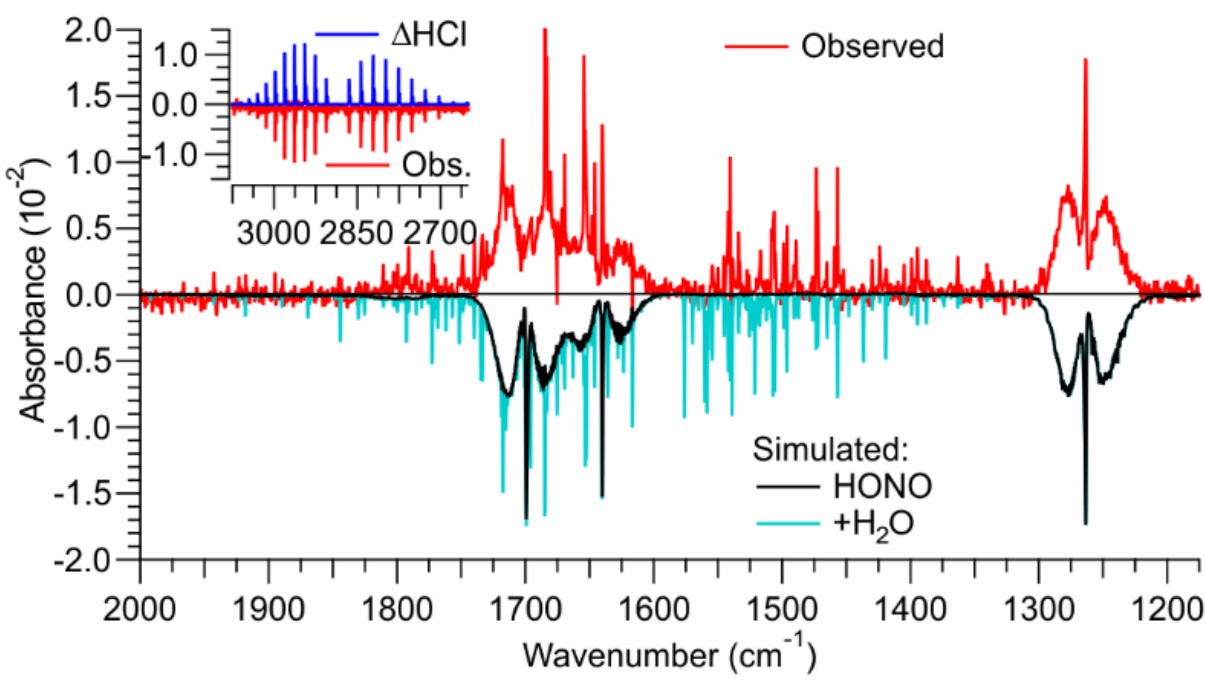

**Figure 2.** Infrared spectrum (shown in red colour) of a gas stream containing HONO generated by reaction of $HCl_{(g)}$ with $NaNO_{2(s)}$ after the source was optimized. The reference spectrum was collected when $NaNO_2$ was bypassed, i.e., contained HCl. Literature spectra (Sharpe et al., 2004) were multiplied by the optical path length of 6.4 m and mixing ratios of the identified trace gases until they reproduced the observed spectrum. The optimized source delivered 3.0 ppmv of HONO from >2.5 ppmv of HCl. The spectrum also contained ~4.0 ppmv of $H_2O$. The HCl and $H_2O$ concentrations are underestimates of their true concentrations since their absorption lines are narrower than the resolution of the FTIR of 0.5 $cm^{-1}$.

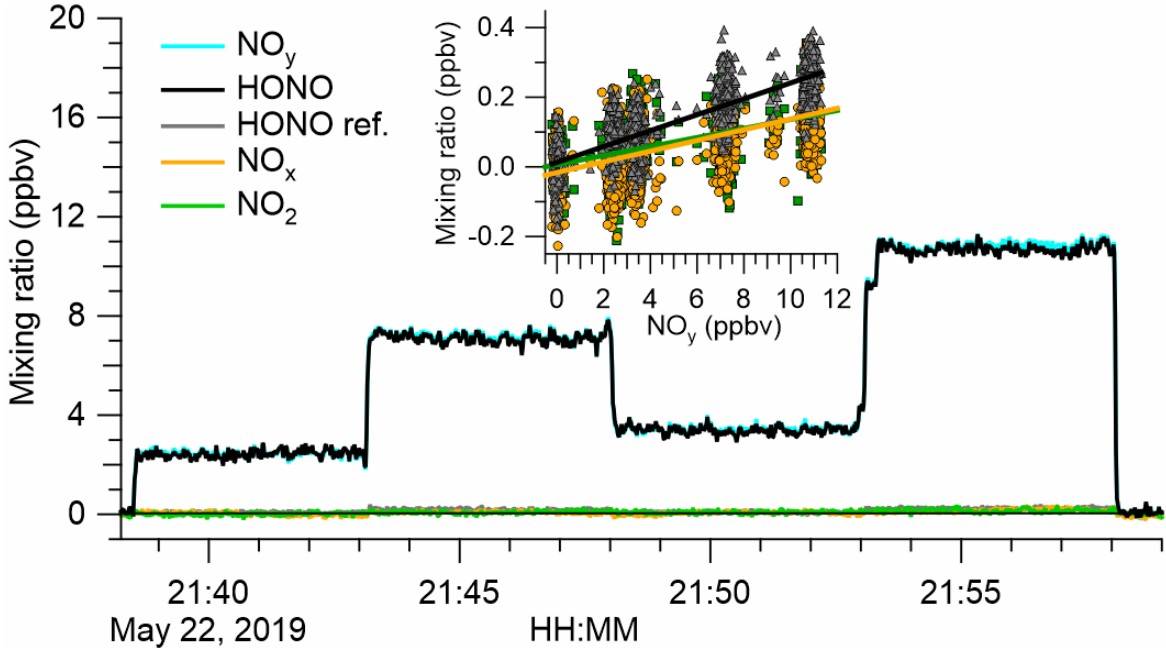

**Figure 3.** Analysis of the HONO source output by TD-CRDS. The TD-CRDS sampled scrubbed "zero" air before 21:38 and after 21:58. The HONO source output was varied by incrementally opening (or closing) the bypass valve. The HONO mixing ratio was calculated by subtracting the response of the "HONO reference" from the $NO_y$ channel. The insert shows scatter plots of $NO_2$, $NO_x$ and "HONO ref" against $NO_y$. Slopes of $(1.29\pm0.06)\%$ for $NO_2$ (points shown in green), $(1.54\pm0.06)\%$ for $NO_x$ (data points shown in orange) and $(2.28\pm0.04)\%$ for "HONO ref") (points shown in grey) were determined, respectively.

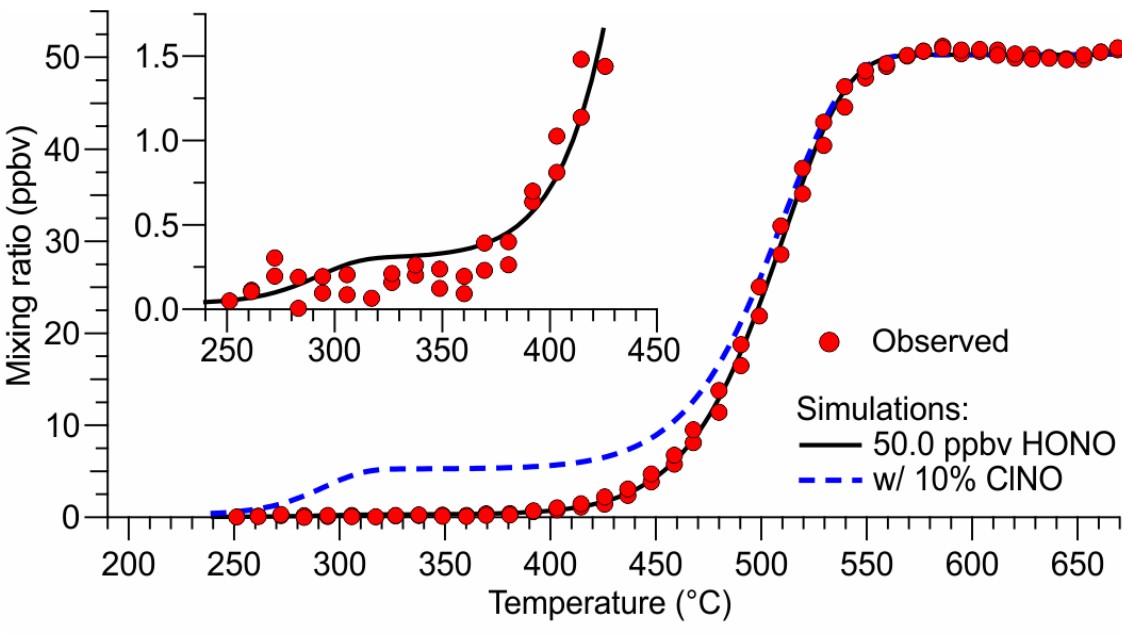

**Figure 4.** Inlet temperature scan when the TD-CRDS was sampling a constant concentration of HONO (data points shown in red). The TD profile was reproduced by a fit to equation (2) with mixing ratios of (49.97±0.18) ppbv, (0.27±0.34) ppbv and (0.03±0.30) ppbv for HONO, ClNO and $NO_x$, respectively (black line). The blue dashed line shows the predicted TD curve of a hypothetical mixture containing 5.0 ppbv ClNO and 45.0 ppbv HONO. The insert shows a close-up of the temperature region in which TD of ClNO occurs.

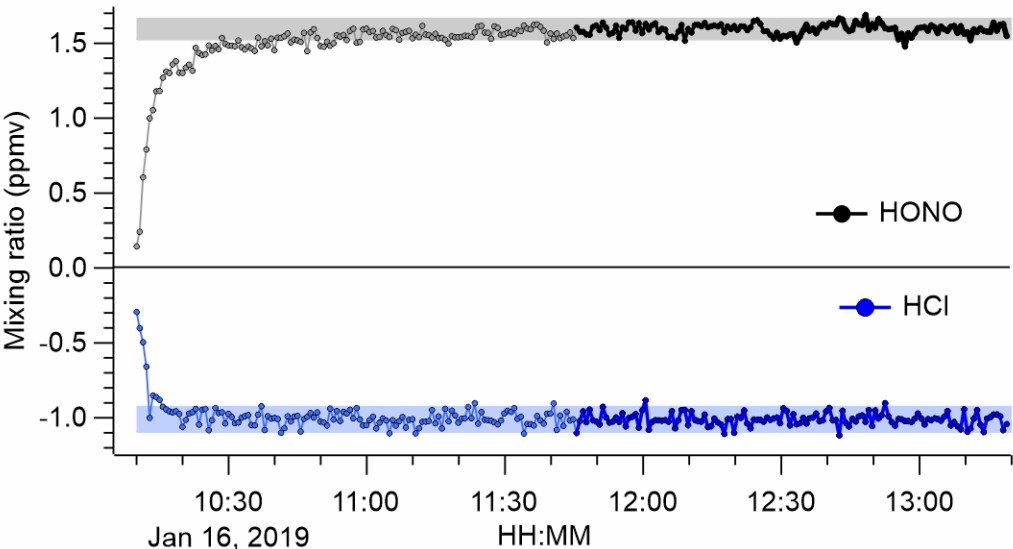

**Figure 5.** Time series of HONO and HCl mixing ratios derived from FTIR analysis of the undiluted HONO
source output. The NaNO$_2$ was placed in line at 10:10. Data acquired during the initial period are shown in
light grey colour. The shaded areas represent the average $\pm$ 2$\sigma$ after the source output was judged to have
stabilized (after 11:45; data points shown in black colour). The 1$\sigma$ precision of the HONO data was
$\pm$35 ppbv and that of the HCl data was $\pm$42 ppbv. The HCl mixing ratios are an underestimate because the
widths of their absorption lines are less than the FTIR's resolution of 0.5 cm$^{-1}$.

410

**Table 1.** Selected HONO generation systems described in the literature. n/d = not disclosed

| Reference | Method | Analytical method | Result | Notes |
|---|---|---|---|---|
| (King and Moule, 1962; Stockwell and Calvert, 1978) | (R1) | UV absorption | mixture of NO, NO$_2$ and HONO | Equilibration of (R1) |
| (Cox, 1974) | (R1) | NO and "total NO$_x$" CL; HONO scrubbed w/ NaOH(0.1N) | mixture of NO, NO$_2$ and HONO | Equilibration of (R1) |
| (Cox and Derwent, 1976) | (R2) | NO and "total NO$_x$" CL; HONO scrubbed w/ NaOH(0.1N) | mixture of NO, NO$_2$ and ~50% HONO | Volatilization of "nitrous fumes" from solution of sulfuric acid and sodium nitrite |
| (Braman and De la Cantera, 1986) | (R3) | NO/NO$_y$ CL analyser (heated Au tube) | mixture of NO, NO$_2$ and 50% - 90% HONO | Sublimation of oxalic acid on sodium nitrite |
| (Taira and Kanda, 1990) | (R2) | NO/NO$_y$ CL analyser (carbon converter); HONO collected on Na$_2$CO$_3$ filters and analysed by IC | mixture containing 2% - 3% NO and 4% - 6% NO$_2$ | Volatilization of HONO from dynamically mixed sodium nitrite and sulfuric acid solution |
| (Febo et al., 1995; Stutz et al., 2000) | (R4) | DOAS (NO$_2$, HONO); NO CL | >99.5% purity | 1 L bath, 2 m reverse diffusion tube, heated stirring reactor |
| (Brust et al., 2000) | (R4) | FTIR (NO$_2$, HONO), IC (HONO) | up to 22ppm; <2% NO$_2$ | 20 cm reverse diffusion, heated reactor |
| (Schiller et al., 2001) | (R4) | TLDAS (HONO), FTIR (HONO) IC of KOH solutions | n/d | bubbled N$_2$ through 0.5 M HCl followed by reaction with solid sodium nitrite |
| (Perez et al., 2007) | (R4) | TD-CL, TD-LIF | observed ClNO | same setup as Febo et al. (1995) |

| | | | | |
|---|---|---|---|---|
| (Perez et al., 2007) | (R2) | TD-CL, TD-LIF | >95% | flowed $H_2SO_4$ aerosol over $NaNO_2$ sandwiched between paper filters |
| (Roberts et al., 2010) | (R4) | IBBCEAS, $NO_y$-CL, CIMS | >95% | HCl gas cylinder (10 ppmv) and $NaNO_2$ reactor tube |
| (Ren et al., 2010) | (R4) | LOPAP, $NO_y$-CL, CIMS | n/d | Reverse diffusion tube, high [HCl] (9-12 M), tubing thickness not described |
| (Reed et al., 2016) | (R2) | Differential photolysis NO-CL; $NO_2$ CAPS; FTIR | HONO (70.4%; < 50 ppbv); NO (15%); $NO_2$ (12.8%); $HNO_3$ (1.3%) | Perm tube filled with 37% HCl placed in oven with $NaNO_2$ salt; flushed @ 1.5 slpm; <50 ppbv |

**Table 2.** Summary of TD-CRDS analyses of the HONO source output. The RH of the diluent gas was in the range of 15% to 35%, except for experiments conducted with cylinder gases which are shown below the dashed line. The range of mixing ratios stated is for output after dilution. The stated errors are from regression analyses of plots of $NO_2$, $NO_x$ or $NO_x$+ClNO vs. $NO_y$ and are at the $1\sigma$ level. n/d = not determined.

| Date | Setting | Diluent gas | Range (ppbv) | HONO (%) | $NO_2$ (%) | $NO_x$ (%) | $NO_x$+ClNO (%) |
|------|---------|-------------|--------------|----------|------------|------------|-----------------|
| Jan 16, 2019 | Lab | Scrubbed air | 0 - 140 | *97.2±0.1 | n/d | 2.80±0.07 | n/d |
| May 3, 2019 | Lab | Scrubbed air | 0 - 7 | 96.0±0.2 | 1.9±0.2 | 4.0±0.2 | (4.0±0.2)** |
| May 6, 2019 | Lab | Scrubbed air | 0 - 93 | 96.4±0.4 | 1.9±0.4 | 3.6±0.4 | (3.6±0.4)** |
| May 10, 2019 | Lab | Scrubbed air | 0 - 21 | *97.6±0.1 | 1.10±0.03 | 2.40±0.05 | n/d |
| May 17, 2019 | Lab | Scrubbed air | 0 - 38 | 98.7±0.1 | 0.01±0.02 | 0.92±0.01 | 1.25±0.01 |
| May 22, 2019 | Lab | Scrubbed air | 0 - 14 | 97.7±0.1 | 1.29±0.06 | 1.54±0.06 | 2.28±0.04 |
| Jan 17, 2020 | Field | Scrubbed air | 0 - 80 | 98.5±0.1 | 1.00±0.05 | 1.50±0.08 | (1.50±0.08)** |
| Mar 19, 2020 | Lab | Scrubbed air | 0 - 320 | *98.72±0.01 | n/d | 1.28±0.01 | n/d |
| Jul 25, 2019 | Lab | Cylinder $O_2$ | 0 - 95 | 86.60±0.02 | 4.60±0.03 | n/d | 13.4±0.02 |
| Jan 30, 2020 | Field | Cylinder $N_2$ | 0 - 50 | 81.3±0.1 | 5.90±0.09 | n/d | 18.7±0.1 |

* Upper limit. ** The TD-CRDS quartz inlet temperature was ramped between 300 °C and 600 °C; the TD-CRDS mixing ratios at an inlet temperature of 300 °C matched the $NO_x$ mixing ratios observed with the room temperature inlet, indicating the absence of ClNO.