# Peer review of "A compact, high-purity source of HONO validated by Fourier Transform Infrared and Thermal Dissociation Cavity Ring-down Spectroscopy"

_Atmospheric Measurement Techniques, 2020_

## Referee Comment (RC1) · Anonymous Referee #1 · 5 Jun 2020

This paper discusses the optimization of a HONO calibration source suitable for field work. For many instruments, the source needs to be relatively free of compounds that could interfere with the measurements of HONO. Current methods of HONO production often produce concentrations that vary with operating conditions (humidity, temperature, flow rates), as well as requiring long periods before stable outputs are achieved. Because of issues of stability and reproducibility, the output of HONO calibration methods need to be confirmed in the field with a secondary HONO instrument, often an absolute measurement technique such as absorbance spectroscopy or a chemilumi-

nescent NOx analyzer. Ideally, a field calibration source would provide stable and reproducible concentrations of HONO that could be quantified in the laboratory and reliably reproduced in the field, such that a secondary instrument would not be needed for source quantification.

The source described in this paper is based on the design of Febo et al. (1995), which has been used extensively by other groups as outlined in the manuscript. The authors describe changes in the design of the source that appear to minimize impurities such as ClNO and NOx, and characterize the output of the source using FTIR and TD-CRDS instruments. They find that their "optimized source" can produce concentrations of HONO with approximately 97% purity and is stable after 1.5 hours. The authors report HONO concentrations from the source ranging between 1.5 and 3 ppm from FTIR experiments and 11 and 50 ppb from TD-CRDS experiments (with some dilution occurring).

The paper does provide some new information regarding the production of a stable, high purity HONO source for instrument calibrations and is likely of interest to the atmospheric chemistry community. While worthy of eventual publication, the paper does not provide sufficient information to give the reader confidence in reproducing the measurements. More experimental details regarding the individual flow rates, humidity, and temperatures should be provided. It is also not clear whether the source routinely produces concentrations of HONO with the stated purity over a range of operating conditions. The authors should include measurements of the HONO concentrations and impurities produced by the source over a range of temperatures and humidities that may be encountered when the source is used in the field. The authors should also provide information on the reproducibility of the source using the same operating conditions after it has been turn off. Illustrating that the output of the source is stable and reproducible from day-to-day might suggest that the source could be robust enough to be used for instrument calibrations without requiring a secondary instrument for quantification in the field.

Specific comments:

Methods: As mentioned in the paper, previous studies have demonstrated that the production of HONO by reaction R4 requires the gas stream to be humidified. However, the authors do not state the humidity of the gas stream used in their experiments.

Page 4, line 88: Is there any reason for the 20 ccm dilution with O2? If so, why O2 and not N2, zero air, or scrubbed room air?

Page 4, last paragraph: How much dilution was typical? What are final flow rates? More experimental details should be provided.

Fig. 1: The position of the needle valve and dilution do not match the written description. The figure only shows the input of the TD-CRDS being diluted and not the FTIR, while the text describes dilution before both instruments. The schematic would benefit with typical flow rates.

Page 7, line 161: In this discussion, the authors are probably referring to reaction R5, and not R6.

Page 7, line 169: It appears that the improvement in source 2 is only a reduction of ClNO contamination, as both the NO and NO2 concentrations increased. This should be clarified.

Page 8, equation 2: Check the equation – there appears to be an error translating the fonts.

Fig. 5: The authors should define the lighter shaded points in both plots. Are the lighter shades indicating points while the source is stabilizing? If so, why are there lighter shade points at the end of the time series? The authors should clarify how they determined that the output had stabilized.

---

## Referee Comment (RC2) · Anonymous Referee #2 · 12 Jun 2020

The authors present an optimized HONO calibration source that is based on the design of Febo et al., 1995, where HCl vapor is passed over solid NaNO2. They used a permeation tube to achieve lower (< 4 ppmv) HCl mixing ratios which is key to achieve higher purity as ClNO formation is slowed down. They quantified the impurities by using FTIR spectroscopy and TD-CRDS and found the optimized design of the source to be of > 97 % purity. With the optimized design (lower HCl supply) they were able to generate HONO concentrations in the low ppm range and by further dilution in the lower ppb range. The stabilization time of 1.5 h was short compared to other source designs.

[Figure]

Furthermore, the source is portable and after stabilization, HONO mixing ratios are readily tunable. Due to its instability, there are no permeation tubes or standard gases for HONO available and it must be produced in situ. Therefore, to calibrate gas phase mixing of HONO such a compact and easy deployable source is of interest for the atmospheric chemistry community. The study is well performed and the manuscript well written. Therefore, I support publication after considering the minor comments given below.

General comments:

As surfaces are unavoidable in laboratory setups, the role of surface reactions should be discussed. Although I guess that HONO formation from NO2 impurities is not of importance, there are also heterogeneous decomposition reactions for HONO that form NO (and NO2) and might therefore important to keep impurities low. See esp. (Finlayson-Pitts et al., 2003).

Regarding the FTIR measurements: Can the authors provide more details about the reference spectrum used for HONO (spectrum of cis or trans isomer or total spectrum? taken at which temperature?). The spectral features will change with temperature as the amounts of cis and trans isomers of HONO change with temperature (e.g. Barney et al., 2000). Furthermore, please provide temperature and humidity values of the gas stream if possible.

Specific comment:

L88: Why diluting with 20 mL min-1 flow of oxygen?

References:

Barney, W. S., Wingen, L. M., Lakin, M. J., Brauers, T., Stutz, J. and Finlayson-Pitts, B. J.: Infrared Absorption Cross-Section Measurements for Nitrous Acid (HONO) at Room Temperature, J. Phys. Chem. A, 104, 1692–1699, 2000.

Febo, A., Perrino, C. and Sparapani, M. Gherardi. R.: Evaluation of a High-Purity and
High-Stability Continuous Generation System for Nitrous Acid, Environ. Sci. Techno, 29, 2390–2395, 1995.

Finlayson-Pitts, B. J., Wingen, L. M., Sumner, A. L., Syomin, D. and Ramazan, K. A.: The heterogeneous hydrolysis of NO2 in laboratory systems and in outdoor and indoor atmospheres: An integrated mechanism, Phys. Chem. Chem. Phys., 5, 223-242, 2003.
* * *

---

## Author Comment (AC1) · 29 Jun 2020

We thank the reviewer for her/his time and comments, which are reproduced in *black italic font* below. Our responses are shown in regular font. Revised text (as it appears in the manuscript) is shown in blue font. Line numbers are those of the revised manuscript with changes accepted.

*Anonymous Referee #1

*This paper discusses the optimization of a HONO calibration source suitable for field work. For many instruments, the source needs to be relatively free of compounds that could interfere with the measurements of HONO. Current methods of HONO production often produce concentrations that vary with operating conditions (humidity, temperature, flow rates), as well as requiring long periods before stable outputs are achieved. Because of issues of stability and reproducibility, the output of HONO calibration methods need to be confirmed in the field with a secondary HONO instrument, often an absolute measurement technique such as absorbance spectroscopy or a chemiluminescent NOx analyzer. Ideally, a field calibration source would provide stable and reproducible concentrations of HONO that could be quantified in the laboratory and reliably reproduced in the field, such that a secondary instrument would not be needed for source quantification.*
*The source described in this paper is based on the design of Febo et al. (1995), which has been used extensively by other groups as outlined in the manuscript. The authors describe changes in the design of the source that appear to minimize impurities such as ClNO and NOx, and characterize the output of the source using FTIR and TDCRDS instruments. They find that their "optimized source" can produce concentrations of HONO with approximately 97% purity and is stable after 1.5 hours. The authors report HONO concentrations from the source ranging between 1.5 and 3 ppm from FTIR experiments and 11 and 50 ppb from TD-CRDS experiments (with some dilution occurring). The paper does provide some new information regarding the production of a stable, high purity HONO source for instrument calibrations and is likely of interest to the atmospheric chemistry community. While worthy of eventual publication, the paper does not provide sufficient information to give the reader confidence in reproducing the measurements. More experimental details regarding the individual flow rates, humidity, and temperatures should be provided. It is also not clear whether the source routinely produces concentrations of HONO with the stated purity over a range of operating conditions. The authors should include measurements of the HONO concentrations and impurities produced by the source over a range of temperatures and humidities that may be encountered when the source is used in the field. The authors should also provide information on the reproducibility of the source using the same operating conditions after it has been turn off. Illustrating that the output of the source is stable and reproducible from day-to-day might suggest that the source could be robust enough to be used for instrument calibrations without requiring a secondary instrument for quantification in the field.*

Response: We appreciate the reviewer's point of view and agree that the manuscript would be strengthened by including a summary of our results with the optimized source in both the lab and the field.

We added the following on line 152 (to the materials and methods section):

"2.4 Field deployment
The HONO source was utilized during the "Study of nitrogen oxides in winter downwind from oil and gas sands" (SNOWDOGS) field campaign in Fort McKay, Alberta, Canada, in January 2020. Its output was quantified in parallel by TD-CRDS and a Thermo 42i-Y NO-NO$_y$ CL instrument equipped

with a Mo converter heated to 325 °C. This converter and the TD-CRDS quartz inlets were mounted on the roof of a trailer which housed the instruments and the HONO source. The TD-CRDS sampled at a flow rate of ~1.2 slpm per channel through ~5 cm short, 300 μm i.d. stainless steel flow restrictors placed inline after the heated quartz sections and before the Teflon filter assembly. The HONO source output was delivered via a 5 m long, 1/4" (0.64 cm) o.d. and 3/16" (0.48 cm) i.d. FEP Teflon tube to both instruments."

We expanded the text on line 232:

"**3.3 Source stability**
The source output gradually decreased over a time scale of weeks of continuous use, which was rationalized by the visible depletion of the HCl permeation tube. However, the source output remained stable and reproducible on shorter time scales. An example FTIR time series is shown in Figure 5, which was acquired after 1 month of intermittent use. After a 1.5 hours stabilization period, the source produced a stable output of 1.57 ppmv of HONO (from >1.0 ppmv of HCl) with a precision of ±35 ppbv.

**3.4 Source purity and day-to-day reproducibility**

The optimized source routinely delivered HONO in high purity (range 96.0% - 98.7%), as long as scrubbed, moisture-containing room air was used as diluent gas (Table 2). When dry cylinder gas was used instead, the source output was stable but contained a larger amount of undesired side products; for example, on July 25, 2019, the output contained 4.6% $NO_2$ and 8.8% NO + ClNO (Table 2). When used in the field, the performance was similar: with scrubbed air, the HONO purity was 98.5%, whereas with dry cylinder gas, the purity was merely 81.3% (Table 2 and Figure S5)."

We added Table 2, which summaries our analysis results, on line 414:

"**Table 2.** Summary of TD-CRDS analyses of the HONO source output. The RH of the diluent gas was in the range of 15% to 35%, except for experiments conducted with cylinder gases which are shown below the dashed line. The range of mixing ratios stated is for output after dilution. The stated errors are from regression analyses of plots of $NO_2$, $NO_x$ or $NO_x$+ClNO vs. $NO_y$ and are at the 1σ level. n/d = not determined.

| Date | Setting | Diluent gas | Range (ppbv) | HONO (%) | NO₂ (%) | NOₓ (%) | NOₓ+ClNO (%) |
|------|---------|-------------|--------------|----------|---------|---------|--------------|
| Jan 16, 2019 | Lab | Scrubbed air | 0 - 140 | *97.2±0.1 | n/d | 2.80±0.07 | n/d |
| May 3, 2019 | Lab | Scrubbed air | 0 - 7 | 96.0±0.2 | 1.9±0.2 | 4.0±0.2 | (4.0±0.2)** |
| May 6, 2019 | Lab | Scrubbed air | 0 - 93 | 96.4±0.4 | 1.9±0.4 | 3.6±0.4 | (3.6±0.4)** |
| May 10, 2019 | Lab | Scrubbed air | 0 - 21 | *97.6±0.1 | 1.10±0.03 | 2.40±0.05 | n/d |
| May 17, 2019 | Lab | Scrubbed air | 0 - 38 | 98.7±0.1 | 0.01±0.02 | 0.92±0.01 | 1.25±0.01 |
| May 22, 2019 | Lab | Scrubbed air | 0 - 14 | 97.7±0.1 | 1.29±0.06 | 1.54±0.06 | 2.28±0.04 |
| Jan 17, 2020 | Field | Scrubbed air | 0 - 80 | 98.5±0.1 | 1.00±0.05 | 1.50±0.08 | (1.50±0.08)** |
| Mar 19, 2020 | Lab | Scrubbed air | 0 - 320 | *98.72±0.01 | n/d | 1.28±0.01 | n/d |
| Jul 25, 2019 | Lab | Cylinder O₂ | 0 - 95 | 86.60±0.02 | 4.60±0.03 | n/d | 13.4±0.02 |
| Jan 30, 2020 | Field | Cylinder N₂ | 0 - 50 | 81.3±0.1 | 5.90±0.09 | n/d | 18.7±0.1 |

* Upper limit. ** The TD-CRDS quartz inlet temperature was ramped between 300 °C and 600 °C; the TD-CRDS mixing ratios at an inlet temperature of 300 °C matched the $NO_x$ mixing ratios observed with the room temperature inlet, indicating the absence of ClNO."

To the SI, we added

[Figure]

"**Figure S5.** Sample field analysis of the HONO source output. The TD-CRDS and CL instruments sampled scrubbed "zero" air before 21:05 and after 21:36. The HONO source output was added at 21:05:30. Only the CL $NO_y$ responded because the TD-CRDS quartz inlet temperature was 200 °C. At 21:10, the temperature of the quartz inlet was increased from 200 °C to 600 °C to quantify HONO. The absence of an inflection point and prior agreement with the $NO_x$ measurement implies the absence of ClNO. At 21:16 and every ~5 min thereafter, the HONO output concentration was decreased by slightly opening the bypass valve. The insert shows scatter plots of $NO_2$ and $NO_x$ (calculated by adding CL NO and CRDS $NO_2$ data) against $NO_y$. Slopes and offsets were $(0.96\pm0.05)$% and $-(30\pm3)$ pptv for $NO_2$ (points shown in green) and $(1.46\pm0.08)$% and $-(164\pm4)$ pptv for $NO_x$ (data points shown in orange), respectively."

Finally, we revised the statement given in the abstract on line 15:

"The source produces gas streams containing HONO in air in > 95%  purity relative to other nitrogen oxides." to account for the observation of $(96\pm0.2)$% purity on May 3, 2019 (Table 2).

*Specific comments:*

*Methods: As mentioned in the paper, previous studies have demonstrated that the production of HONO by reaction R4 requires the gas stream to be humidified. However, the authors do not state the humidity of the gas stream used in their experiments.*

Response: We added the following on line 88: "For the experiments shown in this manuscript, the relative humidity of the diluent gas stream was in the range of 15% to 35%. "

*Page 4, line 88: Is there any reason for the 20 ccm dilution with O2?*

Response: The additional flow is probably not essential, but we found it practical to keep part of the setup under flow of a clean gas - our laboratory air contains surprisingly high levels of $NO_x$ delivered from the building's air intake, plus the continued flow would remove any impurities that might build up. The use of a second mass flow controller was also advantageous as it allowed the residence time and HCl concentration to be changed on the fly without disturbing the permeation tube setup (which can take quite a long time to return to a stable output when flows are changed).

We modified the text on line 90 as follows:

"The permeation tube was placed inside a glass chamber (VICI Dynacalibrator Model 120) whose temperature was controlled at 25.0 °C and which was continuously flushed at a flow rate of ~0.15 L $min^{-1}$ with room air scrubbed using activated charcoal for a period of several days into a fume hood. The use of scrubbed air ensured that the gas stream contained water to maintain efficient HONO production (Schiller et al., 2001).

The HCl output was diluted in a gas stream of $O_2$ (~20 mL $min^{-1}$); this gas stream also served to continuously flush the connecting tubing when not in use."

*If so, why O2 and not N2, zero air, or scrubbed room air?*

Response: The reviewer is correct that either $N_2$, zero air or scrubber air could have been used. We chose $O_2$ because an $O_2$ cylinder was already in place for the $O_3$ generator. No changes were made.

*Page 4, last paragraph: How much dilution was typical? What are final flow rates? More experimental details should be provided.*

Response: We diluted the output by factors between 5 and 200, depending on how much HONO we desired to deliver on any particular day at a flow rate marginally larger than the TD-CRDS inlet sample flow. We modified the paragraph as follows:

"To deliver HONO in atmospheric concentrations (i.e., < 10 ppbv), a portion of the source output was directed towards waste with the aid of a pump and a needle valve. The output concentration could be rapidly changed (typically by factors between 5 to 200) by adjusting the position of the needle valve. The remaining output was diluted using scrubbed air and directed towards the instruments at a final flow rate slightly larger than the sampling requirement of the instruments. "

*Fig. 1: The position of the needle valve and dilution do not match the written description. The figure only shows the input of the TD-CRDS being diluted and not the FTIR, while the text describes dilution before both instruments.*

Response: As stated on line 107, the needle valve was used to deliver HONO in atmospheric concentrations (i.e., < 10 ppbv). The FTIR sampled the output without dilution, and the Figure is accurate as shown. We clarified this on line 111:

"Gas streams exiting the HONO sources (prior to dilution) were analysed using an FTIR spectrometer"

*The schematic would benefit with typical flow rates.*

Response: This information has been added to Figure 1.

*Page 7, line 161: In this discussion, the authors are probably referring to reaction R5, and not R6.*

Response: We thank the reviewer for catching this error. It has been corrected (line 177). We also corrected another error of this type on line 165.

*Page 7, line 169: It appears that the improvement in source 2 is only a reduction of ClNO contamination, as both the NO and NO2 concentrations increased. This should be clarified.*

Response: The reviewer's observation is correct. The issue with both source 1 and source 2 was that there was too much HONO produced, which can generate side products by either R-1 or R5. In source 2, more HONO was produced than in source 1 (because the rate of R5 was reduced) which increased the rate of NO and $NO_2$ produced via R-1.

We added the following on line 185: "... an improvement over source 1 in terms of unwanted ClNO production but still inadequate."

*Page 8, equation 2: Check the equation – there appears to be an error translating the fonts.*

Response: Apologies - this was an error converting word (.docx) to pdf format. Equation (2) is

$$[\text{Observed}]_{\text{total}} = [NO_x] + [ClNO]\left(1-e^{-A_{ClNO}\times e^{\frac{-E_{A,ClNO}}{RT}}\times t_{res}}\right) + [HONO]\left(1-e^{-A_{HONO}\times e^{\frac{-E_{A,HONO}}{RT}}\times t_{res}}\right)$$

This has been corrected in the revised document.

*Fig. 5: The authors should define the lighter shaded points in both plots. Are the lighter shades indicating points while the source is stabilizing? If so, why are there lighter shade points at the end of the time series?*

Response: Our apologies - the two lighter shade points at the end of the time series should have been in dark shade as well; this has been corrected. We also modified the caption as requested.

**"Figure 5.** Time series of HONO and HCl mixing ratios derived from FTIR analysis of the undiluted HONO source output. The $NaNO_2$ was placed in line at 10:10. Data acquired during the initial period are shown in light grey colour. The shaded areas represent the average $\pm$ 2σ after the source output had stabilized (after 11:45; data points shown in black colour). The 1σ precision of the HONO data was $\pm$35 ppbv and that of the HCl data was $\pm$42 ppbv. The HCl mixing ratios are an underestimate because the widths of their absorption lines are less than the FTIR's resolution of 0.5 $cm^{-1}$."

*The authors should clarify how they determined that the output had stabilized.*

Response: In the caption of Figure 5, we had stated "The shaded areas represent the average $\pm$ 2σ after the source output had stabilized (after 11:45; data points shown in black colour)."

Since we didn't apply a mathematically rigorous criterion to determine output stability, we changed the above to:

"The shaded areas represent the average $\pm$ 2σ after the source output  was judged to have stabilized (after 11:45; data points shown in black colour)."

We had judged whether the output was stable (or not) by visual inspection but also had performed a few simple statistical tests for corroboration.

A stable output is achieved when the output mixing ratio is not changing. For the data shown in black, this criterion is met since the slope of a linear regression fit is $-(0.4\pm1.9)\times10^{-6}$ $s^{-1} \approx 0$.

We also calculated the average (μ) and standard deviation (σ) for this period which were 1.596 and $\pm$0.036 ppmv. The residuals during this time period were normally distributed (i.e., reproduced by a Gaussian equation, see figure below).

The μ±2σ interval is shown as a grey area in Figure 5; since the noise is normally distributed, this interval is expected to encompass 95% of the data; for the data shown in black, 6 of the 132 (~4.5%) are outside this confidence interval.

[Figure]

The above confirms that the output during the chosen period was stable (i.e., confirms that "the source output had stabilized" as we had stated in the caption of Figure 5), but it does not justify the choice of the left limit to what is called a stable period in Figure 5 (which we had estimated at 11:45:30).

Expanding on the above discussion, we applied a double exponential fit to the HONO data in Figure 5:

[Figure]

$$(1.595\pm0.004)+(1.28\pm0.04)\times e^{-t/(157\pm9)}+(0.25\pm0.02)\times e^{-t/(1751\pm216)}$$

The first time constant of this fit was $(157\pm9)$ s, corresponding to the mixing time of gas in the FTIR multipass cell (assuming plug flow 450 cm$^3$ at a flow rate of 170 cm$^3$ min$^{-1}$ ~ 160 s). The second time constant was $29.3\pm3.6$ min. The fit is within 1% of the limiting mean at 11:46:56 in Figure 5, which is very close to the "eye-ball" estimate.

---

## Author Comment (AC2) · 29 Jun 2020

We thank the reviewer for her/his time and comments, which are reproduced in *black italic font* below. Our responses are shown in regular font. Revised text (as it appears in the manuscript) is shown in blue font. Line numbers are those of the revised manuscript with changes accepted.

*Anonymous Referee #2

*The authors present an optimized HONO calibration source that is based on the design of Febo et al., 1995, where HCl vapor is passed over solid NaNO2. They used a permeation tube to achieve lower (< 4 ppmv) HCl mixing ratios which is key to achieve higher purity as ClNO formation is slowed down. They quantified the impurities by using FTIR spectroscopy and TD-CRDS and found the optimized design of the source to be of > 97 % purity. With the optimized design (lower HCl supply) they were able to generate HONO concentrations in the low ppm range and by further dilution in the lower ppb range. The stabilization time of 1.5 h was short compared to other source designs.*

*Furthermore, the source is portable and after stabilization, HONO mixing ratios are readily tunable. Due to its instability, there are no permeation tubes or standard gases for HONO available and it must be produced in situ. Therefore, to calibrate gas phase mixing of HONO such a compact and easy deployable source is of interest for the atmospheric chemistry community. The study is well performed and the manuscript well written. Therefore, I support publication after considering the minor comments given below.*

Response: We appreciate the reviewer's supportive comments.

*General comments:*
*As surfaces are unavoidable in laboratory setups, the role of surface reactions should be discussed. Although I guess that HONO formation from NO2 impurities is not of importance, there are also heterogeneous decomposition reactions for HONO that form NO (and NO2) and might therefore important to keep impurities low. See esp. (Finlayson-Pitts et al., 2003).*

Response: We agree with the reviewer that surface chemistry is an important consideration. Finlayson-Pitts et al. (2003) discussed the mechanism of heterogeneous conversion of $NO_2$ (to HONO and $HNO_3$) on Pyrex which involves formation and wall-adsorption of $NO_2$ dimer (i.e., $N_2O_4$) as the initial steps. Another paper by the same group (Syomin and Finlayson-Pitts, 2003) discusses HONO decomposition on glass surfaces, which was linked to wall-adsorbed HONO reacting with wall-adsorbed $HNO_3$.

We believe this chemistry is limited in our setup as we have taken several steps to avoid it becoming factor. First of all, the chemistry described by Finlayson-Pitts et al. (2003) and Syomin and Finlayson-Pitts (2003) was observed in a static system on a time scale of hours or required cell conditioning with $HNO_3$; in our system, the 50 mL reaction vessel was continuously flushed (at a flow of ~150 mL min$^{-1}$ from the zero air generator when in use and continuously with ~20 mL min$^{-1}$ from the $O_2$ cylinder), such that there was little time for this chemistry to create much of an effect. In addition, formation of $N_2O_4$ scales with the square of the $NO_2$ concentration, which was low in our experiments. More importantly, the Pyrex reaction vessel was heated, which promotes partitioning of wall-adsorbed species to the gas-phase and reduces the extent of surface chemistry. Furthermore, the equilibrium

constant for the reaction $2NO_2 \rightleftharpoons N_2O_4$ is temperature-dependent favoring $N_2O_4$ dissociation at higher temperature. The NASA-JPL compilation recommends an equilibrium constant of $5.9 \times 10^{-29}$ cm$^3$ molecule$^{-1}$ e$^{(6643/T)}$ (Burkholder et al., 2015). When the temperature is increased from room temperature to 50 °C, the equilibrium constant decreases by a factor of ~6.

The following was added to the manuscript on line 97:

"In the second source ("source 2"), the NaNO$_2$ was placed in a two-neck, 50 mL Pyrex round bottom flask which was covered in aluminium foil to prevent photolysis of nitrite and nitrous acid and was externally heated to a temperature of 50 °C using a water bath and mechanically agitated using a magnetic stir bar as described by Febo et al. (1995). Heating this vessel promotes dissociation of molecular clusters such as N$_2$O$_4$ and partitioning of wall-adsorbed molecules to the gas-phase, which, if present, can drive HONO decomposition on borosilicate glass surfaces (Syomin and Finlayson-Pitts, 2003)."

Another consideration is surface chemistry in the connecting tubing. We constructed all our connecting tubing out of inert Teflon material and kept it under continuous flow (with an inert gas such as $N_2$ or $O_2$) when not in use (a detail we added to the manuscript - see our response to the question on L88 below).

*Regarding the FTIR measurements: Can the authors provide more details about the reference spectrum used for HONO (spectrum of cis or trans isomer or total spectrum? taken at which temperature?). The spectral features will change with temperature as the amounts of cis and trans isomers of HONO change with temperature (e.g. Barney et al., 2000). Furthermore, please provide temperature and humidity values of the gas stream if possible.*

Response: The reference spectrum was obtained from the Pacific Northwest National Laboratory digital library (Sharpe et al., 2004). According to their meta data file, they acquired this spectrum at 25 °C and atmospheric pressure and with dry nitrogen as diluent gas and generated HONO in situ from the reaction of NaNO$_2$ with HCl. We would hence expect the same distribution of HONO isomers as we would have generated in our experiments, which were conducted at room temperature.

We added more detail to section 2.2 ("Analysis of HONO source output by FTIR") on line 115 as requested by the reviewer:

"Room-temperature spectra were acquired continuously at a time resolution of 30 s. …. Mixing ratios of trace gases were determined from fits (by least squares error minimization over a selected wavelength range) of room-temperature reference spectra from the Pacific Northwest National Laboratory (Sharpe et al., 2004), …. the pressure inside the multi-pass cell (which was equal to that of the room) was monitored using a pressure transducer (Omegadyne PX419–015A5V)."

Because the bulk of the diluent gas was generated by passing room air through a scrubber, the relative humidity of the gas was approximately the same as the room (~30%; accounting for dilution with dry oxygen and neglecting H$_2$O co-emitted by the HCl perm tube, RH is estimated at ~25% for the data shown in Figure 2). Since the background spectrum was acquired with this H$_2$O background, the FTIR only shows drifts in H$_2$O. The following was added on line 187:

"A sample FTIR spectrum of this source's output is shown in Figure 2. With a freshly prepared HCl permeation tube and an estimated relative humidity in the round bottom flask of ~25%, > ~2.5 ppmv HCl were consumed and ~3.0 ppmv of HONO were produced. "

*Specific comment:*
*L88: Why diluting with 20 mL min-1 flow of oxygen?*

Response: The additional flow is probably not essential, but we found it practical to keep part of the setup under flow of a clean gas - our laboratory air contains surprisingly high levels of $NO_x$ delivered from the building's air intake, plus the continued flow would remove any impurities that might build up. The use of a second mass flow controller was also advantageous as it allowed the residence time and HCl concentration to be changed on the fly without disturbing the permeation tube setup (which can take quite a long time to return to a stable output when flows are changed).

We modified the text as follows:

"The permeation tube was placed inside a glass chamber (VICI Dynacalibrator Model 120) whose temperature was controlled at 25.0 °C and which was continuously flushed at a flow rate of ~0.15 L $min^{-1}$ with room air scrubbed using activated charcoal for a period of several days into a fume hood. The use of scrubbed air ensured that the gas stream contained water to maintain efficient HONO production (Schiller et al., 2001). The HCl output was diluted in a gas stream of $O_2$ (~20 mL $min^{-1}$); this gas stream also served to continuously flush the connecting tubing when not in use."

*References:*
*Barney, W. S., Wingen, L. M., Lakin, M. J., Brauers, T., Stutz, J. and Finlayson-Pitts, B. J.: Infrared Absorption Cross-Section Measurements for Nitrous Acid (HONO) at Room Temperature, J. Phys. Chem. A, 104, 1692–1699, 2000.*
*Febo, A., Perrino, C. and Sparapani, M. Gherardi. R.: Evaluation of a High-Purity and High-Stability Continuous Generation System for Nitrous Acid, Environ. Sci. Technol, 29, 2390–2395, 1995.*
*Finlayson-Pitts, B. J., Wingen, L. M., Sumner, A. L., Syomin, D. and Ramazan, K. A.: The heterogeneous hydrolysis of NO2 in laboratory systems and in outdoor and indoor atmospheres: An integrated mechanism, Phys. Chem. Chem. Phys., 5, 223-242, 2003.*
Burkholder, J. B., Sander, S. P., Abbatt, J. P. D., Barker, J. R., Huie, R. E., Kolb, C. E., Kurylo, M. J., Orkin, V. L., Wilmouth, D. M., and Wine, P. H.: Chemical Kinetics and Photochemical Data for Use in Atmospheric Studies, Evaluation Number 18, National Aeronautics and Space Administration, Jet Propulsion Laboratory, California Institute of Technology, Pasadena, California, 2015.
Schiller, C. L., Locquiao, S., Johnson, T. J., and Harris, G. W.: Atmospheric measurements of HONO by tunable diode laser absorption spectroscopy, J. Atmos. Chem., 40, 275-293, 10.1023/A:1012264601306, 2001.
Sharpe, S. W., Johnson, T. J., Sams, R. L., Chu, P. M., Rhoderick, G. C., and Johnson, P. A.: Gas-phase databases for quantitative infrared spectroscopy, Appl. Spectrosc., 58, 1452-1461, 2004.
Syomin, D. A., and Finlayson-Pitts, B. J.: HONO decomposition on borosilicate glass surfaces: implications for environmental chamber studies and field experiments, Phys. Chem. Chem. Phys., 5, 5236-5242, 10.1039/b309851f, 2003.